# Early Clinical Outcomes of the First Commercialized Human Autologous Ex Vivo Cultivated Oral Mucosal Epithelial Cell Transplantation for Limbal Stem Cell Deficiency: Two Case Reports and Literature Review

**DOI:** 10.3390/ijms24108926

**Published:** 2023-05-18

**Authors:** Hiroshi Toshida, Tomoto Kasahara, Masamichi Kiriyama, Yuma Iwasaki, Jobu Sugita, Kohei Ichikawa, Toshihiko Ohta, Katsumi Miyahara

**Affiliations:** 1Department of Ophthalmology, Juntendo University Shizuoka Hospital, Shizuoka 410-2211, Japan; t.kasahara.pu@juntendo.ac.jp (T.K.); m.kiriyama.ci@juntendo.ac.jp (M.K.); y.iwasaki.dl@juntendo.ac.jp (Y.I.); jsugita@juntendo.ac.jp (J.S.); k-ichika@juntendo.ac.jp (K.I.); izuganka@ninus.ocn.ne.jp (T.O.); 2Laboratory of Morphology and Image Analysis, Biomedical Research Core Facilities, Juntendo University Graduate School of Medicine, Tokyo 113-0033, Japan; katsuo@juntendo.ac.jp

**Keywords:** limbal stem cell deficiency, cultivated oral mucosal epithelial cell transplantation, stem cell, regeneration therapy, COMET, Ocural^®^

## Abstract

The first product in the world for ex vivo cultivated oral mucosal epithelial cell transplantation (COMET) to treat limbal stem cell deficiency (LSCD), named Ocural^®^, was launched in June 2021 in Japan. COMET was performed on two patients, including the first case in the post-marketing phase of Ocural^®^. Pathological and immunohistochemical examinations were also carried out using specimens obtained before and after COMET and the spare cell sheet. In case 1, the ocular surface remained free from epithelial defects for approximately six months. In case 2, although defect of the cornea-like epithelia was observed after COMET for one month, it was resolved after the insertion of lacrimal punctal plugs. In case 1, adjuvant treatment was interrupted due to an accident during the second month after COMET, resulting in conjunctival ingrowth and corneal opacity. Eventually, a lamellar keratoplasty was required at six months after COMET. Immunohistochemistry revealed the presence of markers for stem cells (p63, p75), proliferation (Ki-67), and differentiation (Keratin-3, -4, and -13) in both the cornea-like tissue after COMET and a cultivated oral mucosal epithelial cell sheet. In conclusion, Ocural^®^ can be accomplished without major complications, and the stem cells derived from oral mucosa might be successfully engrafted.

## 1. Introduction

Limbal stem cell deficiency (LSCD) is a refractory ocular surface disease characterized by a loss of limbal stem cells, which are vital for repopulation of the corneal epithelium [1]. As a result of this loss, conjunctivalization of the cornea occurs, contributing to severe loss of corneal clarity and serious impairment of visual function. Conventional treatments for LSCD include allogeneic limbal transplantation and keratoepithelioplasty [2,3], but they are associated with a risk of graft rejection and infectious keratitis [4,5,6,7]. Amniotic membrane has also been used for the treatment of LSCD [8].

In 1997, Pellegrini et al. reported a new technique that used ex vivo cultivated autologous limbal epithelial cells for transplantation [9]. In this technique, limbal epithelial cells harvested from a healthy limbal area of a patient are cultivated ex vivo to fabricate a cell sheet for transplantation. This new treatment was the first in the world to be approved and commercialized, under the name Holoclar^®^ in Europe in 2019 [10,11,12]. However, whether its etiology is genetic (e.g., aniridia), acquired (e.g., Stevens–Johnson syndrome, ocular cicatricial pemphigoid, trauma, or thermal burns), or idiopathic, LSCD is often a bilateral disease in which no, or very small, healthy limbal areas are available, making autologous limbal epithelial cell transplantation impracticable and invasive [2]. Therefore, newer techniques beyond limbal cells were sought. The oral mucosal epithelium emerged as a promising source of transplant tissue because it is a stratified squamous epithelium, such as the corneal epithelium, and has similar cellular characteristics [13,14,15]. In 2004, Nakamura et al. first described a newer technique in which autologous oral mucosal epithelial cells were cultivated on amniotic membrane to create a cell sheet for transplantation onto the ocular surface in human [16]. This technique has since been referred to as ex vivo cultivated oral mucosal epithelial cell transplantation (COMET) and has been extensively studied in many countries for corneal surface reconstruction in patients with LSCD [16,17,18,19,20,21,22,23,24,25,26,27,28,29,30,31,32,33,34,35,36,37,38,39,40,41,42,43,44].

Several challenges remained before a COMET-based regenerative medicine product could be commercialized, including how to minimize damage to the cell sheet caused during enzymatic treatment for detachment and how to enhance graft adhesion to the host corneal stroma. In an attempt to find an answer to the former challenge, Nishida et al. studied corneal reconstruction with epithelial cell sheets created by ex vivo cultivation of autologous cells on temperature-responsive cell-culture surfaces, a cultivation technique that could bypass the need for enzymatic sheet treatment [17,45]. This first report of the use of a temperature-responsive cell-culture dish in ophthalmology described its application in COMET. On 11 June 2021, the first-ever COMET product for the treatment of LSCD, named Ocural^®^, was approved as a regenerative medicine product by the Pharmaceuticals and Medical Devices Agency, the regulatory agency in Japan responsible for the approval and supervision of pharmaceuticals and medical devices [46]. However, the first regenerative medicine product manufactured with this temperature-responsive cultivation technique for ophthalmic use, which was approved in Japan, was a non-COMET, human autologous limbal epithelial cell sheet product [46,47]. Nepic^®^ was launched on 19 March 2020, specifically for the treatment of unilateral LSCD. On the other hand, Ocural^®^ is indicated for bilateral LSCD. Ocural^®^ is composed of a cultivated oral mucosal epithelium package and a tissue transportation set consisting of tubes used to transport oral mucosal tissue, harvested at a medical institution, and blood [47,48]. The second COMET product, Sakracy^®^, was introduced into the Japanese market on 20 January 2022. It embodies the above-mentioned technique of the first COMET report with human autologous oral mucosal epithelial cells cultivated on human amniotic membrane and is indicated for the relief of symblepharon in patients with LSCD [16].

Thus, as at the end of 2022, two COMET products have been commercialized in Japan, making Japan the first country where this is the case. In Japan, Holoclar^®^ has not yet been released. Table 1 summarizes the characteristics of the three products. In Japan, these products for corneal surface reconstruction were listed in the National Health Insurance (NHI) tariff in April 2022, enabling patients with LSCD to use these health services under NHI coverage. Since then, at our hospital we have used Ocural^®^ in two patients with LSCD and followed them up for six months and four months, respectively. Herein, we present our experience with these patients and discuss the advantages of the product, perioperative precautions, and issues in postoperative management. The characteristics of the transplanted cells were also examined by immunohistochemistry.

## 2. Results

### 2.1. Clinical Results

#### 2.1.1. Case 1

The left eye of a 79-year-old man with LSCD developed gradual corneal opacity over 12 years after he received lamellar keratoplasties (LKPs) twice due to iatrogenic corneal perforation during corneal scraping for removal of corneal plaques. His left eye showed corneal opacity with prominent conjunctival encroachment and corneal neovascularization (Figure 1), leading to a diagnosis of stage III LSCD. His right eye was atrophied, and the right eye had total corneal opacity, indicating a diagnosis of phthisis bulbi. So, he had bilateral uncorrectable visual acuity loss, with visual acuity in the right eye at the counting fingers (CF) level and a visual acuity in the left eye at the light perception level. Because his left eye appeared to be an indication for Ocural^®^, he was scheduled to undergo the operation after giving informed consent. The result of Schirmer’s tear test (STT) I was 13 mm on the left eye. The patient’s previous systemic illnesses included mitral valve insufficiency, arrhythmia, and cerebral infarction.

After four weeks of cultivation, the oral mucosal epithelial cells had expanded and reached the required cell density for COMET. Oral mucosal epithelial sheet transplantation to the surface of eye was accomplished without complications. The specimens obtained during the surgery were served for H&E staining and immunohistochemical staining. Postoperatively, the patient received a three-day course of oral antibiotic therapy and a corticosteroid at 20 mg/day for one week and then at a dose tapered down at a rate of 5 mg per week. Topical treatments included levofloxacin 1.5% eye drops (Santen Pharmaceutical Co., Ltd., Osaka, Japan) instilled four times a day for two weeks and betamethasone sodium phosphate 0.1% eye drops (Shionfogi & Co., Ltd., Osaka, Japan) instilled four times a day for an indefinite period of time. A therapeutic contact lens (CL) (Air Optix^®^ EX Aqua, Alcon Co., Ltd., Fort Worth, TX, USA) was replaced every week and continued for one month. The immediate postoperative outcome was very favorable, and no corneal epithelial loss was seen (Figure 2A,B), including anterior segment optical coherence tomography (Figure 2C). However, in the second postoperative month, the patient fell at home and injured his leg, which prevented his follow-up in the subsequent three months. During this time, the only treatment the patient received was artificial lacrimal fluid purchased from a pharmacy at his discretion. At a follow-up visit five months after surgery, the patient’s left eye showed corneal opacity (Figure 2D), which was an indication for LKP. On the other hand, there were still few corneal epithelial defects (Figure 2E). At six months after COMET (Figure 2F), the patient underwent LKP and vascular cauterization. The specimens obtained during the surgery were served for hematoxylin and eosin hematoxylin and eosin (H&E) staining and immunohistochemical staining.

After LKP, corneal clarity was regained (Figure 3), and visual acuity had improved slightly to the HM level at one month after surgery.

#### 2.1.2. Case 2

A 52-year-old man with LSCD had developed cerebral infarction four years before treatment with COMET and had since had persistent lagophthalmos, which eventually led to bilateral corneal opacity, conjunctivalization, and neovascularization (Figure 4). His medical history also included surgical resection of a pterygium in the left eye performed 30 years earlier. At the first presentation, the visual acuity of the right eye was 0.4 (corrected visual acuity, 0.7), and the visual acuity of the left eye was at the HM level. The right eye was diagnosed as having stage IA LSCD; central corneal clarity was maintained only in areas 2 mm and 1 mm lateral and medial of the areas of conjunctivalization and neovascularization of the cornea. The left eye was diagnosed as having stage III LSCD with full conjunctivalization and severe neovascularization of the cornea. The fundus of the right eye remained normal, but the fundus of the left eye was invisible.

The condition of the left eye appeared to be an indication for COMET and was treated by this procedure at four weeks after the patient’s oral mucosal tissue was harvested as in case 1. Therapeutic CL was worn and replaced every week. The patient received a three-day course of oral antibiotic therapy and a corticosteroid at 20 mg/day for one week and then at a dose tapered down at a rate of 5 mg per week. Topical treatments included levofloxacin eye drops instilled four times a day for two weeks and betamethasone sodium phosphate 0.1% eye drops and sodium hyaluronate 0.1% eye drops instilled four times a day for an indefinite period of time. As of two weeks after COMET, the left eye showed corneal epithelial defect (Figure 5A,D). Although CLs were replaced weekly, corneal epithelial defect persisted (Figure 5B,E). We reviewed the medical record and confirmed that the STT value was 0 mm tested before the surgery. It was resolved within two weeks after the insertion of lacrimal punctal plugs (Eagle Vision^®^ SuperFlex^®^ Punctum Plugs, Hilco Vision, Mansfield, MA, USA), and the visual acuity of the left eye has improved slightly to the CF level. Since then, up to the present time, which is four months after the surgery, there has been no recurrence of the corneal epithelial defect (Figure 5C,F).

### 2.2. Results of Histological and Immunohistological Studies

#### 2.2.1. Histological Study

The specimens were examined with H&E staining (Figure 6A–C). In Figure 6A, the corneal epithelium exhibited irregularities as the corneal epithelium appeared to have increased layers in case 1. The histological images of the epithelial tissue obtained after COMET surgery showed thicker layers of epithelium and a stratification of the basal layer compared to corneal epithelium before COMET in case 1 (Figure 6B). The histological image of the cultured oral mucosal epithelial cell sheet taken from case 2 showed several layers of epithelial cells (Figure 6C).

#### 2.2.2. Immunohistochemistry of Keratins and Muc5AC

The pattern of expression of the keratins and Muc5AC were investigated with immunohistochemistry. Negative control sections had no specific activity in the epithelial regions incubated with normal mouse and rabbit IgG. Cornea-specific keratin-3 was expressed sporadically throughout the entire layer of the epithelia in the specimens taken from case 1 before COMET (Figure 7A), cultivated oral mucosal sheet of case 2 (Figure 7C) and after COMET except basal layer (Figure 7B). Cornea-specific keratin-12 was expressed only in corneal tissues before COMET (Figure 7D). In contrast, keratin-12 was not expressed in any layers of the tissues after COMET (Figure 7E) and cultivated oral epithelial sheet (Figure 7F). Keratin-4 and -13, markers of cornea, conjunctiva, and oral mucosa, were expressed throughout the entire layer in the corneal tissue before COMET (Figure 7G,J) and post-COMET (Figure 7H,K). In contrast, the immunoactivity was different between keratin-4 and -13 in the oral mucosal sheet. Keratin-4 displayed reactivity throughout the entire epithelial layer (Figure 7I), whereas keratin-13 was not expressed in the basal layer of the epithelium (Figure 7L). Muc5AC, a marker of conjunctival goblet cells [21,49], was not observed in the specimens obtained before or after COMET in case 1 (Figure 7M,N), as well as in the cultivated oral epithelial sheet from case 2 (Figure 7O).

#### 2.2.3. Immunohistochemistry of p63, p75, and Ki-67

p63, a transcription factor putative epithelial stem cell marker [17,23,42,43,49,50], was expressed in all specimens (Figure 8A–C). The immunoreactivity of p63 was stronger and its density was higher in the basal cell layers in the post-COMET sample (Figure 8B). In the oral mucosal sheet, p63 showed reactivity in the basal cell layer (Figure 8C). p75, a marker for stem/progenitor cells of oral epithelium [24,31,43,51] and limbal epithelial cells [52,53], was also expressed in all specimens (Figure 8D–F). The immunoreactivity of p75 was stronger and its density was higher in the basal cell layers in the post-COMET sample (Figure 8E). In the oral mucosal sheet from case 2, p75 showed reactivity mainly in the basal cell layer (Figure 8F). Ki-67, a marker of actively cycling cells [31,51], was expressed in the basal layer in all specimens (Figure 8G–I). Ki-67 showed reactivity in a few basal cells before COMET (Figure 8G) and the oral mucosal sheet (Figure 8I). In contrast, the density of Ki-67-positive cells was higher in the post-COMET specimen (Figure 8H) than others.

## 3. Discussion

Regenerative medicine techniques, such as limbal epithelial cell sheet transplantation and COMET, have been studied extensively to explore their use for corneal regeneration to treat ocular surface diseases, such as LSCD [16,17,18,19,20,21,22,23,24,25,26,27,28,29,30,31,32,33,34,35,36,37,38,39,40,41,42,43,44,45,46,47,48]. The approval and commercialization of the first human somatic stem cell product for ophthalmic use, in 2015, was Holoclar^®^. Following that, Nepic^®^ and Ocural^®^ were released in Japan after a government-controlled clinical trial [46,47]. The study adhered to the stipulations of good clinical practice within a well-defined clinical protocol, and strict quality control measures were implemented for the fabrication of cell sheets in a facility compliant with good manufacturing practice (GMP) standards. Even after commercialization, these standards have been maintained. They opened a new era of regenerative medicine for ocular surface disease. The prominent characteristics of these regenerative medicine products are culturing of harvested epithelial cells on temperature-responsive cell-culture dishes and the use of a contract cell-culture service provider. The use of temperature-responsive cell-culture dishes considerably reduce the risk of being unable to detach the cultured cells from the culture surfaces and the associated anxiety if the temperature is optimally reduced. Table 1 shows a list of regenerative medicine products for ocular surface reconstruction. It is not a task that can be performed by physicians or co-medical workers in the hospital in their spare time, so the use of a contract cell-culture service provider to create cell sheets for transplantation represents a breakthrough in regenerative medicine [17]. In the two cases described here, we successfully accomplished COMET with the first of the two cell sheets without having to use the second sheet provided as a reserve.

Major graft factors that may compromise the success of COMET include a paucity of donor cells, which may preclude their sufficient expansion to create a cell sheet transplant, and the potential of bacterial, fungal, mycoplasma, or other pathogenic contamination. The patient-related risk factor of the greatest concern is inability to undergo the operation as scheduled for reasons such as acute infections (e.g., COVID-19 and influenza), pyrexia, other acute illnesses, and accidents. If a patient fails to undergo COMET as scheduled, the operation cannot be rescheduled and needs to be canceled. The cost of one COMET procedure amounts to about 10 million US dollars for cell harvesting, culturing, and transportation between the operation room and the culturing facility plus 0.5 million US dollars for cell sheet transplantation; according to the NHI system in Japan, 30% of the cost is to be paid by the patient. Thus, cancellation of a COMET procedure causes enormous economic loss. Fortunately, we have not encountered such cancellations. However, one of our patients (case 1) suffered a leg injury due to a fall, which led to the interruption of postoperative follow-up of the patient, as described.

Clinical signs of the success of COMET are improvements of ocular surface condition and VA. Carbal et al. reviewed the available articles on clinical outcomes of COMET published between 2004 and 2019 and reported that COMET achieved a stable ocular surface in 172/243 eyes (70.8%) and improved VA in 225/331 eyes (68.2%) [54]. Of the cases reported here, the first patient (case 1) had stage III LSCD before COMET, which improved to stage IA afterwards until the second postoperative month before a three-month loss of follow-up due to accidental injury. There was no corneal epithelial erosion observed until the sixth month after the surgery. After LKP, the patient is at risk of developing corneal neovascularization and corneal graft rejection. Because the COMET-treated eye on case 2 also has corneal stromal opacity, penetrating keratoplasty will be needed before long. In case 2, although persistent epithelial defect was observed immediately after the surgery, it improved after insertion of the lacrimal punctal plug. STT value was 0 mm in case 2 instead of 13 mm in case 1. It was suggested, based on the results from only two cases, that the presence of dry eye may be involved in the aggravation of epithelial defects.

Our finding of greatest impact from these early cases is the importance of postoperative treatment for reducing inflammation. Sufficient anti-inflammatory treatment seems to be needed even after COMET, a regenerative technique that uses autologous cells. The surgical procedure per se may induce inflammation in patients with LSCD, in whom the ocular surface is usually in poor condition. Furthermore, there is a room to refine the culturing technique; for example, culturing techniques have been investigated that use only human tissues and cells to create xeno-free transplants [20,43]. Creation of transplants on amniotic membrane or fibrin-coated culture inserts has also been reported [30], and in 2022 the former technique was commercialized as Sakracy^®^ in Japan. The product will be used and assessed for effectiveness at many institutions.

The considerations based on the obtained immunohistochemical staining results are as follows. In the pre-COMET state of LSCD in case 1, corneal-specific Keratin-3 and Keratin-12, as well as conjunctival-specific Keratin-4 extending throughout the entire corneal layer, and Keratin-13 detected in the superficial epithelium, were observed. This suggests that conjunctival invasion and subsequent conjunctivalization occurred in the corneal epithelium in this case. After COMET, keratin-3, keratin-4, and keratin-13 were expressed both in the epithelium and in the oral mucosal sheet. Additionally, corneal-specific Keratin-12 was not detected in either sample. Furthermore, Muc5AC, which is typically associated with conjunctival goblet cells, was not observed in the specimens collected after the COMET. These results suggest that cells derived from the oral mucosal sheet were maintained even six months after COMET. On the other hand, both specimens exhibited the presence of p63 and p75, which are considered stem/progenitor cell markers [17,23,24,31,42,43,49,50,51,52,53]. The strong expression of these markers in the epithelial basal layer after COMET, along with the expression of the cycling cell marker Ki-67 in the epithelial basal layer, suggested that stem cells derived from the oral mucosal sheet were transplanted, promoting cell proliferation and differentiation. This could be considered a positive result, as it may indicate successful stem cell transplantation. However, after COMET in case 1, the cornea experienced stromal and epithelial opacification, possibly due to a three-month interruption of anti-inflammatory treatment, which could have led to thickening of both layers. This means that we must consider the possibility of evaluating a pathological cornea in this case.

The most common postoperative complication after COMET is corneal epithelial loss. According to the above-mentioned review by Cabral et al. [54], this complication occurs in more than a half (52.8%) of eyes receiving COMET; the next most common complications are increased intraocular pressure (15%) and infection (9.4%). In our cases, none of these complications occurred in the very early postoperative phase, although we will continue to follow the patients to monitor for and, if necessary, treat complications. Many previous reports described gradual corneal neovascularization (which is not regarded as a complication) in the periphery of the cornea in most patients after COMET. Sufficient attention should be paid to the possible occurrence of such an event because it may be a sign of protracted inflammation and may affect the subsequent outcome of COMET. The Japanese guidelines for COMET [55] include the following brief recommendations on how to manage adverse reactions: “In harvesting oral mucosal epithelial tissue, the physician should be alert to the possible infection, inflammation, and opening of the biopsy wound and should provide appropriate treatments to liquefactive events. To treat corneal epithelial loss, application of a therapeutic CL and/or ophthalmic ointment should be considered. To treat dry-eye feeling or keratoconjunctivitis sicca, topical application of artificial lacrimal fluid or other eye drops for dry eye, or insertion of punctal plugs should be considered. To treat ocular hypertension or glaucoma, topical or oral use of an intraocular pressure-lowering agent should be considered, and even surgical interventions should sometimes be considered. Infectious keratitis should be diagnosed and treated according to these guidelines. To treat corneal perforation, application of a therapeutic CL should be considered, and if this measure is ineffective, surgical treatment such as keratoplasty should be considered.”

To date, no consensus has been reached regarding postoperative anti-inflammatory management in terms of the choice of drugs, route of administration (systemic or topical), and duration of use. In previously reported clinical applications of COMET, postoperative medical treatments were primarily comprised of topically applied artificial lacrimal fluid, serum, or hyaluronic acid. The duration of use of topical antibiotics ranged from two weeks to six months. The primary measures for postoperative anti-inflammation were topical and topical/systemic use of corticosteroids, and other postoperative medications included topical cyclosporin [20,27,33,34], cyclophosphamide [16,24], and topical tacrolimus after penetrating keratoplasty that followed COMET [38]. There seems to be an urgent need to establish a standard protocol for postoperative inflammation control after COMET. Individualized anti-inflammatory treatment will be needed for patients with protracted inflammation.

Practitioners who have reported on the clinical use of COMET have shared many aspects of surgical procedures and techniques and postoperative management; however, they have followed their own procedures in harvesting oral mucosal tissue, cultivating oral mucosal epithelial cells (e.g., reagents and culture media used), and, in some aspects, managing postoperative patients. The commercialization of COMET products will result in these treatment aspects becoming more sophisticated, standardized, and updated with new findings. The commercialization of multiple COMET products in Japan marks only the starting point of this corneal reconstructive technique, and evidence will accumulate as findings are reported from many patients treated by this technique. COMET-eligible patients with LSCD often have refractory disease and are likely to develop various adverse reactions at varying frequencies. And as mentioned above, individualized postoperative management will be needed, especially for patients with protracted postoperative inflammation. There is an urgent need to accumulate evidence to address all these issues. The early outcomes of COMET in the two patients reported here appear to indicate that the procedure can be successful. We will further follow these and future patients to identify optimal measures to achieve better postoperative outcomes of COMET.

## 4. Materials and Methods

### 4.1. Indications for COMET

Before the commercialization of COMET and its listing in the NHI tariff, the Japanese Ophthalmological Society and the Keratoplasty Society of Japan organized a working group to establish requirements for the use of human (autologous) oral mucosal epithelial cell sheets. A working group was formed to establish guidelines for the appropriate use of this product (Japanese guidelines, second edition) [55]. As this was originally written in Japanese, it is summarized below.

#### 4.1.1. Choice of Product

Figure 9 shows a flow diagram of the decision process for choosing the appropriate product. Nepic^®^ is indicated for patients with unilateral LSCD, whereas patients with bilateral LSCD need to be treated with COMET; Ocural^®^ should be used in patients without symblepharon, and Sakracy^®^ in those with symblepharon.

#### 4.1.2. Eligible Patients

According to the Japanese guidelines, COMET is indicated for LSCD that exhibits either of the following two patterns: (1) corneal conjunctivalization of at least 50% of the total limbal area that extends to areas 2.5 mm or less from the corneal center in the affected eye; and (2) corneal conjunctivalization that extends to areas 2.5 mm or less from the corneal center after suboptimal response to surgical debridement of conjunctival scarring (and amniotic transplantation, as necessary) (Figure 10). As per the stage classification by Deng et al. [1], COMET is indicated for the treatment of stage IIB or III LSCD. In addition, stage IIA when the removal of the conjunctival scar tissue and, if necessary, amniotic membrane transplantation was performed on the affected eye, but the effect was insufficient and the conjunctivalization extended to the area within 5 mm in diameter including the corneal center of the affected eye.

Nepic^®^ shares these indications and requirements with Ocural^®^ but is different from Ocural^®^ in that it is only indicated for unilateral LSCD and is contraindicated in patients with Stevens–Johnson syndrome, graft-versus-host disease, congenital LSCD (e.g., aniridia), recurrent pterygium, and idiopathic LSCD.

#### 4.1.3. Qualification of Operating Physicians

The Japanese guidelines recommend that physicians who perform COMET should be ophthalmologists with adequate knowledge, expertise, and experience in keratoplasty who meet all of the following requirements:(1)is a member of the Japan Cornea Society and the Keratoplasty Society of Japan and is registered as an ophthalmologist with the Japanese Ophthalmological Society;(2)has performed keratoplasty on at least five eyes; and(3)has completed a training course organized by the product manufacturer.

According to the request by the Japanese regulatory authority, for administrative reasons a physician and not a dentist should harvest the oral mucosal tissue. An otolaryngologist or plastic surgeon can do this work on behalf of the ophthalmologist, but only if he or she has completed the above-mentioned training course because this work requires adequate knowledge of oral mucosal harvesting and post-biopsy management.

### 4.2. Ethics Statement

The study protocol was reviewed and approved by the ethics committee of Juntendo University Shizuoka Hospital, Izunokuni, Japan (approval number: 809). Both patients signed the informed-consent form to participate in the study. In addition, the study protocol complied with the guidelines for human studies and the World Medical Association Declaration of Helsinki.

### 4.3. Surgical Procedures and Techniques

#### 4.3.1. Harvesting of Oral Mucosal Tissue

For successful COMET, patients underwent dental treatment and oral hygiene control. The oral mucosal tissue was harvested with local anesthesia using a modified boat-shaped incision technique and immersed in saline containing antibiotics. The specimen was thoroughly washed with saline and iodine solution and then transported to a cell-culture facility with a tissue transportation set [48].

#### 4.3.2. Culturing and Sheet Fabrication

The specimen was received by the cell-culture facility. Oral mucosal epithelial cells were isolated and, after enzymatic treatment, seeded on 3T3-derived feeder cells and subjected to subcultures. Once sufficient expansion of the epithelial cells on temperature-responsive cell-culture dishes was confirmed, sheet transplantation was scheduled for at least four weeks later [17,47].

#### 4.3.3. Surgical Procedure

The oral mucosal epithelial sheet was received from the cell-culture facility on the day of surgery and was transplanted for limbal stem cell deficiency in the patient. The bulbar conjunctiva covered by the cornea was removed. The sheet was transplanted and sutured onto the living tissue. A therapeutic CL was applied immediately after the surgery [17].

### 4.4. Histological and Immunohistological Studies

All tissues were fixed in 10% buffered formalin, embedded in paraffin, and cut in 3 μm thick sections. All specimens were examined with H&E to confirm localization of tissues and cells. Specimens were immunostained, using the previously described method [56]. Deparaffinized tissue sections were immersed in methanol with 0.3% H_2_O_2_ to block endogenous peroxidase, then flooded with 1% bovine serum albumin, and 1% normal goat serum to minimize nonspecific antibody binding. Sections were incubated with anti-keratin 3, 4, 12, and 13, and anti-p63, anti-p75, and anti-Ki-67 antibodies shown in Table 2 overnight at 4 °C. After washing, sections were incubated for 30 min with peroxidase-labeled polymer-horseradish peroxidase (HRP) conjugated secondary antibodies (Envision + System HRP; DAKO, Carpinteria, CA, USA). After washing with phosphate-buffered saline, development of peroxidase was achieved using freshly prepared 3,3′-diaminobenzidine tetrahydrochloride (Sigma, St. Louis, MO, USA) 25 mg in 100 mL of 0.01 mol/L PBS containing 0.015% hydrogen peroxidase. Negative controls without primary antibodies were include with each experiment.

## 5. Conclusions

Ocural^®^, the world’s first COMET for LSCD, was launched in Japan; early outcomes show successful engraftment, few complications, and in the post-surgery tissue and cell sheet revealed stem cell markers, proliferation, and differentiation.

## Figures and Tables

**Figure 1 ijms-24-08926-f001:**
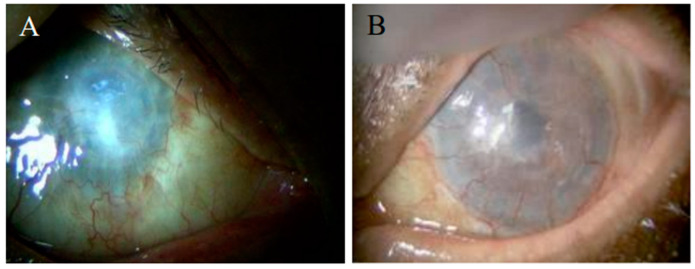
Anterior segment images before COMET in case 1. The right eye exhibited phthisis bulbi (**A**). The left eye displayed corneal opacity with prominent conjunctival encroachment and corneal neovascularization following lamellar keratoplasties (**B**), and was diagnosed as Stage III LSCD.

**Figure 2 ijms-24-08926-f002:**
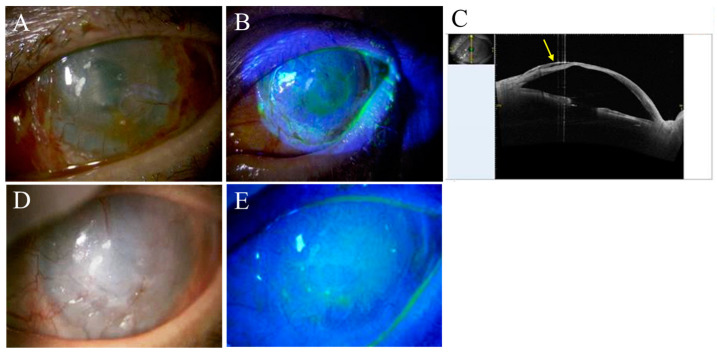
Anterior segment images after COMET on the left eye in case 1. On the fifth postoperative day, the transparency of the cornea had improved (**A**), and there was no staining with fluorescein (**B**), and no epithelial defects were observed. An examination by anterior segment OCT revealed that, although the original corneal stroma was thinning, the corneal epithelium was adhered overall, except for the space between the layers in the arrow region (**C**). After a lapse of three months without visiting the hospital, an overall increase in corneal opacity was observed (**D**) and a few small punctate fluorescein stainings were observed but most of them were covered by the epithelium (**E**) on the sixth postoperative month.

**Figure 3 ijms-24-08926-f003:**
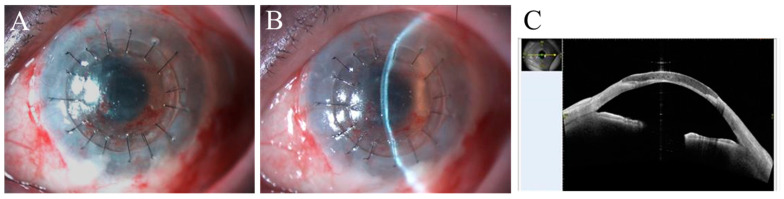
Anterior segment images after LKP on the left eye in case 1. The left eye, which had undergone COMET six months prior, underwent LKP, and, on the third postoperative day, transparency had already been restored (**A**). One week after the surgery, adhesion of the donor cornea to the host cornea was demonstrated by slit-lamp microscopy (**B**) and anterior segment OCT imaging (**C**).

**Figure 4 ijms-24-08926-f004:**
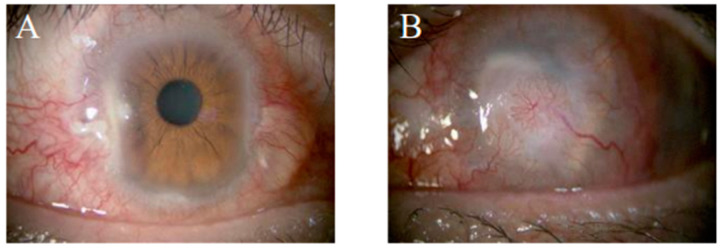
Anterior segment images before COMET in case 2. Bilateral corneal opacity, conjunctivalization, and neovascularization were developed after cerebral infarction since had persistent lagophthalmos. Although the transparency of the central cornea in the right eye was maintained (**A**), the entire cornea of the left eye was opaque, with corneal neovascularization invading from all around, and was diagnosed with LSCD Stage III (**B**).

**Figure 5 ijms-24-08926-f005:**
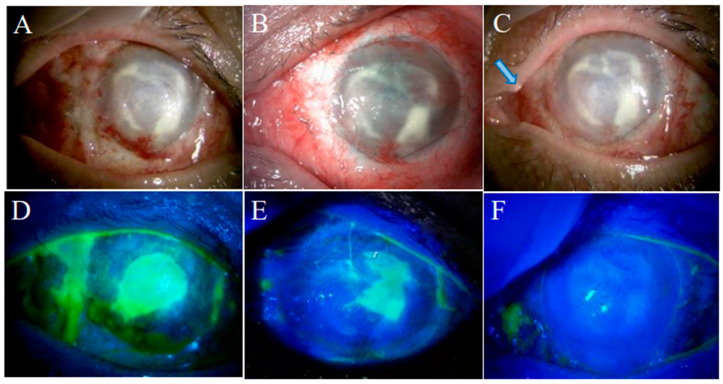
Anterior segment images after COMET on the left eye in case 2 The left eye showed corneal epithelial defect on the second postoperative week (**A**,**D**). As it persisted for one month (**B**,**E**), insertion of lacrimal punctal plugs was conducted. After that, the persistent corneal epithelial defect disappeared within two weeks (**C**,**F**). Arrow shows the head of the punctal plug.

**Figure 6 ijms-24-08926-f006:**
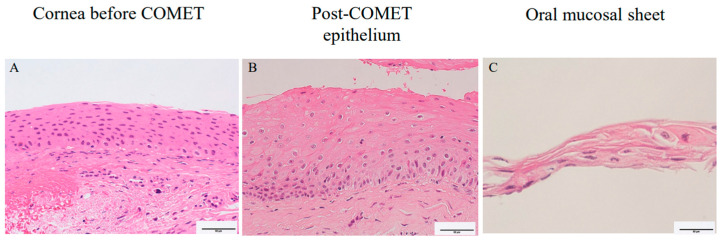
Hematoxylin and Eosin staining of corneal tissue before COMET (**A**) and post-COMET epithelial tissue taken when lamellar keratoplasty was conducted (**B**) in case 1 and spare oral mucosal sheet of case 2 (**C**) stained with hematoxylin and eosin for histological study. Scale bars = 50 μm (**A**,**B**). Scale bars = 20 μm for oral mucosal sheet (**C**).

**Figure 7 ijms-24-08926-f007:**
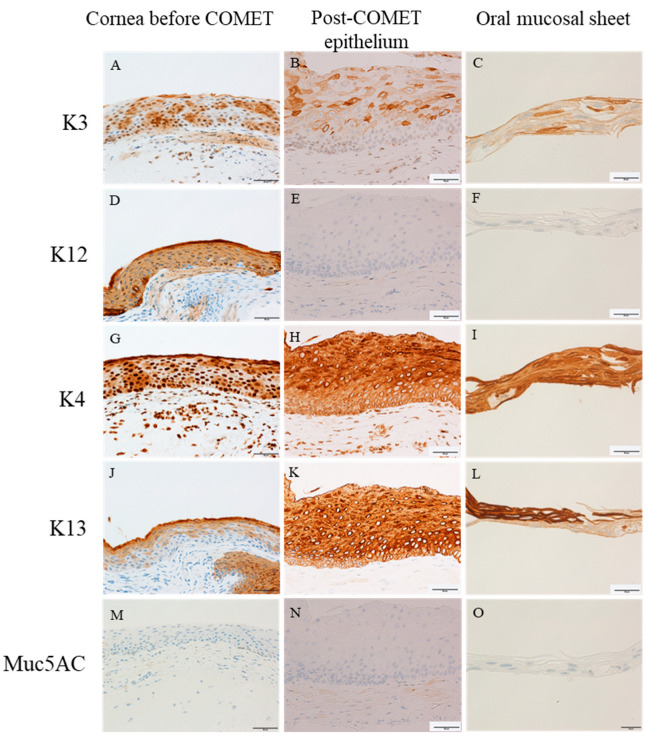
Immunohistochemistry of keratins immunohistochemistry for keratin-3 (**A**–**C**), keratin-12 (**D**–**F**), keratin-4 (**G**,**I**), keratin-13 (**J**–**L**), and Muc5AC (**M**–**O**). Scale bars = 50 μm for corneal tissue before COMET in case 1 (**A**,**D**,**G**,**J**,**M**) and post-COMET epithelial tissue in case 1 (**B**,**E**,**H**,**K**,**N**). Scale bars = 20 μm for oral mucosal sheet of case 2 (**C**,**F**,**I**,**L**,**O**).

**Figure 8 ijms-24-08926-f008:**
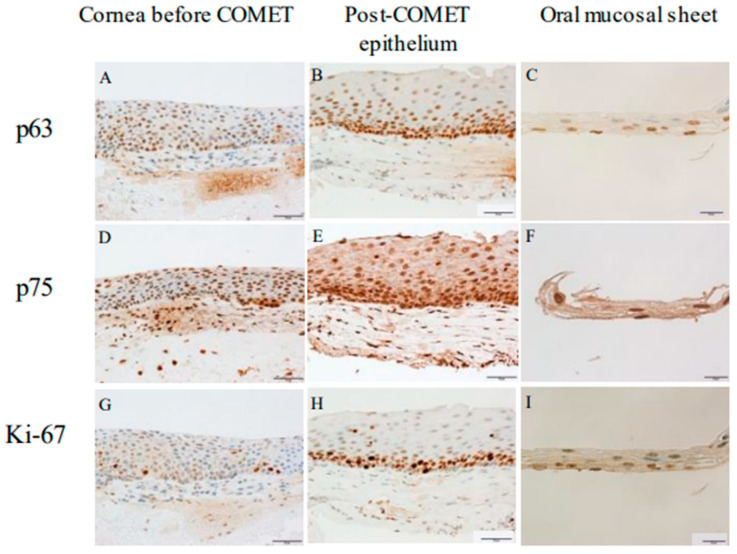
Immunohistochemistry of stem cell markers and actively cycling cell marker. Immunohistochemistry for stem cell markers; p63 (**A**–**C**) and p75 (**D**–**F**), and actively cycling cell marker; Ki-67 (**G**–**I**). Scale bars = 50 μm for corneal tissue before COMET (**A**,**D**,**G**) and post-COMET epithelial tissue in case 1 (**B**,**E**,**H**). Scale bars = 20 μm for oral mucosal sheet of case 2 (**C**,**F**,**I**).

**Figure 9 ijms-24-08926-f009:**
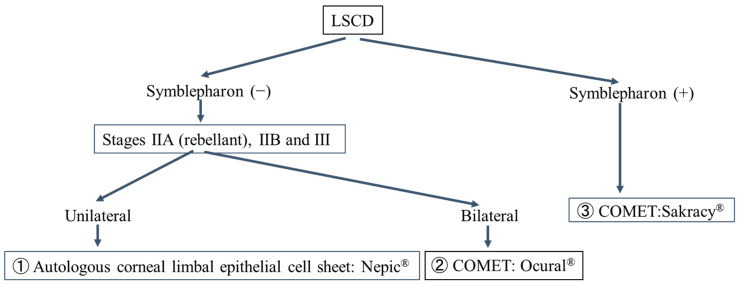
Selections of regenerative medicine products for LSCD (except Holoclar^®^). Indication for Holoclar is different. According to product information, indications are the following: treatment of adult patients with moderate to severe limbal stem cell deficiency (defined by the presence of superficial corneal neovascularization in, at least, two corneal quadrants, with central corneal involvement, and severely impaired visual acuity), unilateral or bilateral, due to physical or chemical ocular burns. A minimum of 1–2 mm^2^ of undamaged limbus is required for biopsy.

**Figure 10 ijms-24-08926-f010:**
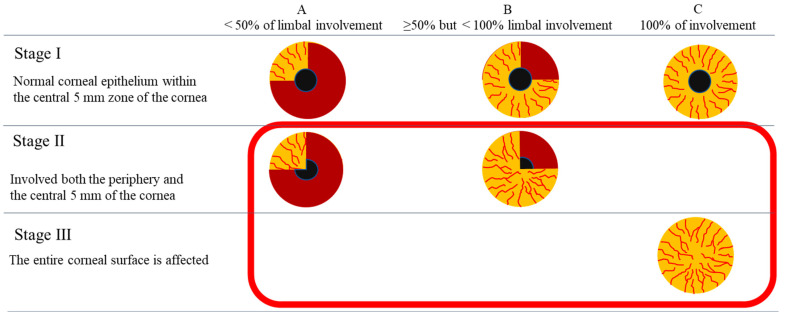
Indications of Nepic^®^ and Ocural^®^ for treatment of LSCD. The indications for Nepic^®^ and Ocural^®^ were following conjunctivalization of the cornea involving ≥50% of the limbus, the same as Stage ⅡB and Ⅲ, and extending to areas.

**Table 1 ijms-24-08926-t001:** List of successful regenerative medicine products commercially agreements for reconstruction of the ocular surface.

Product	Detail of the Product	Indication	Commercial Agreement Companies	Area and Countries	Date of Authorization
Holoclar^®^	Human corneal limbal epithelium	LSCD due to physical or chemical ocular burns	Holostem and Chiesi Farmaceuti	EU	17 February 2015
Nepic^®^	Human corneal limbal epithelium	LSCD–unilateral	J-TEC and Nidek	Japan	19 March 2020
Ocural^®^	Oral mucosal epithelium (COMET)	LSCD–bilateral	J-TEC and Nidek	Japan	11 June 2021
Sakracy^®^	Oral mucosal epithelium with amniotic membrane (COMET)	LSCD with adhesion	Hirosaki Lifescience Innovation	Japan	20 January 2022

**Table 2 ijms-24-08926-t002:** List of Primary Antibodies and Source.

Antibodies	Category	Dilution	Source	Catalog Number
Keratin-3	mouse monoclonal antibody	×100	Santa Cruz Biotechnology, Dallas, TX, USA	sc-80000
Keratin-4	rabbit polyclonal antibody	×200	Proteintech Group, Rosemont, IL, USA	16572-1-AP
Keratin-12	rabbit monoclonal antibody	×500	Abcam, Cambridge, UK	ab185627
Keratin-13	rabbit monoclonal antibody	×100	Abcam	ab92551
Muc5AC	mouse monoclonal antibody	×100	Sigma-Aldrich, St. Louis, MO, USA	ZMS1133
Ki-67	rabbit polyclonal antibody	×8000	Proteintech Group	27309-1-AP
p63	rabbit polyclonal antibody	×400	Proteintech Group	12143-1-AP
p75NTR	rabbit polyclonal antibody	×500	Proteintech Group	55014-1-AP

## Data Availability

The data supporting these findings are available from the corresponding author upon reasonable request.

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
