# Peer review of "Early Clinical Outcomes of the First Commercialized Human Autologous Ex Vivo Cultivated Oral Mucosal Epithelial Cell Transplantation for Limbal Stem Cell Deficiency: Two Case Reports and Literature Review"

_ijms, 2023, doi:10.3390/ijms24108926_

Round 1
Reviewer 1 Report
What are the challenges that need to be addressed before a COMET-based regenerative medicine product can be commercialized?
What is the technique used by Nishida et al. to address one of the challenges in COMET-based regenerative medicine products?
How many COMET products have been commercialized in Japan as of the end of 2022, and what are their names?
What is the significance of the products being listed in the National Health Insurance (NHI) tariff in Japan?
What was the condition of Case 1, and why was the patient scheduled to undergo the Ocural® operation?
Was the transplantation of oral mucosal epithelial sheet successful in Case 1?
How long were the patients in Case 1 followed up, and what were the issues in postoperative management?
What was examined by immunohistochemistry in Case 1?
What is the name of the product launched for the treatment of unilateral LSCD in Japan, and when was it launched?
In what patients is NEPIC® indicated, and what are the contraindications for its use?
What is the stage classification by Deng et al for LSCD, and for what stage is COMET indicated?
What is the minimum requirement for undamaged limbus biopsy for Holoclar treatment?
Can an otolaryngologist or plastic surgeon harvest oral mucosal tissue for COMET treatment, and under what conditions?
What are the qualifications required for physicians who perform COMET according to the Japanese guidelines?
What was the approval number for the study protocol of the COMET treatment, and which hospital's ethics committee approved it?
What guidelines did the study protocol for COMET treatment comply with?
It's ok.
Reviewer 2 Report
Summary
In this study by Toshida et al., two patients with limbal stem cell deficiency (LSCD) who underwent Ocural, an ex vivo cultivated oral mucosal epithelial cell transplantation (COMET) therapy, were evaluated for their post-operative outcomes. The authors found that the engraftment was successful in both patients with few corneal epithelial defects. The defects were resolved in case 2 with the insertion of lacrimal punctal plugs. Both patients had an improvement in the visual acuity of the affected eye to the HM or CF level. The immunohistochemistry results also suggest that there is successful transplantation of oral mucosal epithelial stem cells, as well as maintenance of the cells post-COMET.
Minor Issues
1. The follow-up time in this manuscript after COMET is 4 months for case 2 and 6 months for case 1, which seems to be rather short to evaluate the success of the procedure. Nonetheless, the study setup is sound and the manuscript contributes to the literature on the early clinical outcomes on COMET.
2. Table 1 can be more informative if the authors can also briefly describe the ex vivo cultivation technique for each product, for example Ocural involves cultivating cells on a temperature-responsive cell culture surface.
3. Figures 6, 7, 8: It would be good to state in the figure legend how many weeks/months before and after COMET the corneal tissue was obtained from case 1.
4. Figure 7: K13 staining of the post-COMET epithelium is rather strong. Since K13 is also a marker of conjunctival epithelial cells, it may also be an indicator of conjunctivalisation of the cornea. I’m wondering if the authors can also stain for MUC5AC, which is reported to be a marker for conjunctival goblet cells? The absence of MUC5AC staining may provide stronger evidence for successful engraftment.
5. Figure 8F: I understand from the manuscript that Figures 6, 7, 8 (except 8F) are showing the staining of samples from case 1, so it is a little odd that oral mucosal sheet from case 2 is used for 8F.
6. Table 2: it would also be good to state the catalog or clone number of the antibodies used.
Some examples of minor grammatical and typographical errors are listed here (this list is by no means exhaustive and there may be more errors in the text that I didn’t point out):
- Lines 72 – 73, “This product…” and “Whereas…”: the structure of these 2 sentences is awkward.
- Line 91 and 257, “Rist”: typo error.
- Line 285, “While the results…”: grammatical errors in this sentence.
- Line 373 - 374: awkward sentence structure.
- Line 458: grammatical error.
Author Response
Please see theattachment.
